# Causal contributions of left inferior and medial frontal cortex to semantic and executive control
Sandra Martin [1,3] ✉, Matteo Ferrante[1,3], Andrea Bruera[1] & Gesa Hartwigsen [1,2]

Semantic control enables context-guided retrieval from memory, yet its distinction from domain-general executive control remains debated. We applied transcranial magnetic stimulation (TMS) to the left inferior frontal gyrus (IFG) and pre-supplementary motor area (pre-SMA) to probe their functional relevance for semantic and executive control. Across four sessions, 24 participants received repetitive TMS, followed by semantic fluency, figural fluency, and picture naming tasks. Stimulation of either region broadly disrupted both semantic and figural fluency, suggesting shared functionality. However, electric field modeling of the induced stimulation strength revealed distinct specializations: The left IFG was primarily associated with semantic control, affecting primarily verbal fluency, while the pre-SMA played a domain-general role in executive functions, affecting non-verbal fluency and cognitive flexibility. Notably, only dual-site TMS impaired accuracy in figural fluency, providing unique evidence for successful compensation of executive functions through either the left IFG or pre-SMA following single-site perturbation. These findings underscore the multidimensionality of cognitive control and suggest a flexible contribution of the IFG to control processes, either as semantic-specific or general executive resource. Furthermore, they highlight the tightly interconnected network of executive control subserved by the left IFG and pre-SMA, advancing our understanding of the neural basis of cognitive control.

Semantic cognition—the ability to flexibly access and use knowledge—relies on two key components: semantic memory, which stores concepts and facts, and semantic control, which guides the targeted retrieval based on context. Neuroimaging and lesion studies reveal distinct yet interacting brain networks supporting these processes[1,2]. Semantic memory is represented by a distributed network of modality-specific areas, reflecting all the different information relevant to semantic concepts, and a transmodal hub in the anterior temporal lobe, which bridges information across modalities[3,4]. This network is organized bilaterally, reflecting the broad distribution of conceptual knowledge across the cortex. In contrast, semantic control engages a primarily left-lateralized network involving frontal and temporal regions. The left posterior middle temporal gyrus and the left inferior frontal gyrus (IFG) have emerged as key regions of the semantic control network[2,5,6]. Current debates center on the involvement of domain-specific versus domain-general control networks in semantic processing[7–9]. Here, we address this question by probing the functional relevance of two key regions of semantic and domain-general executive control in verbal and non-verbal fluency tasks.

The involvement of domain-general executive control in semantic processing remains contentious. A recent meta-analysis on the neural correlates of semantic control unveiled a primarily specialized network for semantic control, while also identifying overlapping regions with the domain-general multiple-demand network (MDN), specifically in the pre-supplementary motor area (pre-SMA) and the left IFG[5]. Functional neuroimaging studies have revealed a gradient within these areas, indicating a functional subdivision between semantic-specific and domain-general control regions[7,8,10]. However, studies that explicitly modulated the cognitive and semantic demand of a task showed increased MDN activity under high task loads, suggesting at least a supportive role of the MDN in semantic control[11–13].

Fluency tasks offer a unique opportunity to explore the interplay between domain-specific and domain-general executive control as they rely on executive functions such as processing speed, updating, and inhibition[14,15], while also engaging distinct domain-specific control processes. For example, semantic fluency, a common neuropsychological test, requires the goal-directed navigation of semantic memory for

[1]Research Group Cognition and Plasticity, Max Planck Institute for Human Cognitive and Brain Sciences, Leipzig, Germany. [2]Wilhelm Wundt Institute for Psychology, Leipzig University, Leipzig, Germany. [3]These authors contributed equally: Sandra Martin, Matteo Ferrante. ✉e-mail: martin@cbs.mpg.de

controlled lexical retrieval[16,17]. In contrast, figural fluency, which requires participants to create original line designs, engages visuospatial and visuomotor skills and is considered an important test of non-verbal fluency[18].

In this study, we used transcranial magnetic stimulation (TMS) to investigate the causal roles of the pre-SMA and left anterior IFG in semantic-specific and domain-general executive control. Across four sessions, we applied offline repetitive TMS (rTMS) either to the left anterior IFG alone, to the pre-SMA alone, or subsequently to both regions using a dual-site stimulation approach; a fourth session with sham stimulation served as a control. After each stimulation session, participants completed semantic and figural fluency tasks (see Fig. 1). Comparing the effects of single-site and dual-site TMS allowed us to investigate the potential joint contribution of both areas to fluency performance and their capacity to compensate for perturbation of the other area[19]. We hypothesized that IFG perturbation would selectively impair semantic fluency, while pre-SMA disruption would affect both tasks. Combined stimulation was expected to have a stronger effect on semantic fluency if these regions jointly contribute to the process. Using electrical field (e-field) modeling, we explored the relationship between stimulation strength and behavioral changes, and associated the TMS effect with subregions in the stimulated cortical areas. Additionally, we employed a novel machine learning-based clustering and switching analysis to examine TMS effects on category switching during verbal fluency, a process highly sensitive to the detection of neurodegenerative diseases[20]. Clustering refers to the production of words within a semantic or phonemic subcategory, such as listing several farm animals in a row when asked to name animals. Switching is defined as the ability to shift from one subcategory (cluster) to another, for example, moving from farm animals to zoo animals during the task. Given the executive nature of switching performance, we anticipated a relatively stronger impact of pre-SMA and dual-site TMS compared to IFG stimulation alone.

## Results

We report data from 24 healthy young participants, who each participated in four sessions with rTMS over the left IFG, pre-SMA, dual-site (left IFG succeeded by pre-SMA), and sham stimulation (Fig. 1a, b). After stimulation, participants performed three tasks: verbal, semantic and non-verbal

figural fluency, and picture naming (Fig. 1c). We were interested in modulatory effects of rTMS on reaction times (RTs) and accuracy in each task.

Following recent methodological considerations for TMS studies in cognition[21,22], we included a relatively large sample for a repeated measures design with four sessions and used linear and generalized mixed-effects regression models as well as e-field simulations in our statistical analysis, which allowed us to account for the substantial individual and item-specific variability in response to TMS, while revealing reliable and robust stimulation effects.

### TMS over the left IFG and Pre-SMA delays both semantic and figural fluency

Mixed-effects regression for RTs and accuracy revealed main effects of effective stimulation in all tasks. Specifically, for semantic fluency, all stimulation conditions were associated with slower reactions relative to sham stimulation ($\beta_{IFG} = 0.09$, $p = 0.024$, $\beta_{Pre\text{-}SMA} = 0.12$, $p = 0.007$, $\beta_{Dual\ site} = 0.11$, $p = 0.015$, Fig. 2a and Supplementary Table S2). Stimulation of the pre-SMA induced the relative largest increase in RTs, with an average delay of 189 ms compared to sham. There was no significant difference between the effective stimulation conditions. Results also showed significant main effects for category difficulty with slower reactions for difficult compared to easy categories ($\beta_{Difficult\ categories} = 0.48$, $p < 0.001$, Supplementary Fig. S3a) and slower responses with progressing time ($\beta_{Trial\ index} = 0.002$, $p = 0.025$, Supplementary Fig. S3a). Moreover, there was a significant main effect for executive abilities, where a higher executive score was associated with faster responses ($\beta_{Executive\ abilities} = -0.20$, $p = 0.028$, Supplementary Fig. S3a). Executive abilities were based on an average score of z-standardized scores of the Trail Making Test[23] (TMT) and the Digit Symbol Substitution Test[24] (DSST). We also tested for a potential interaction of stimulation condition and category type, hypothesizing that TMS may selectively show stronger disruptions for more demanding semantic categories. However, the interaction was not significant ($X_{Stimulation\ condition:Category\ type} = 0.70$, $p = 0.873$) and reduced overall model performance compared to the original LMM without the interaction term present ($AIC_{SemFluency\ RT\ Orig} = 37821$, $AIC_{SemFluency\ RT\ Int} = 37837$).

Analyzing the number of correct trials per session revealed no significant effect of stimulation for accuracy in semantic fluency

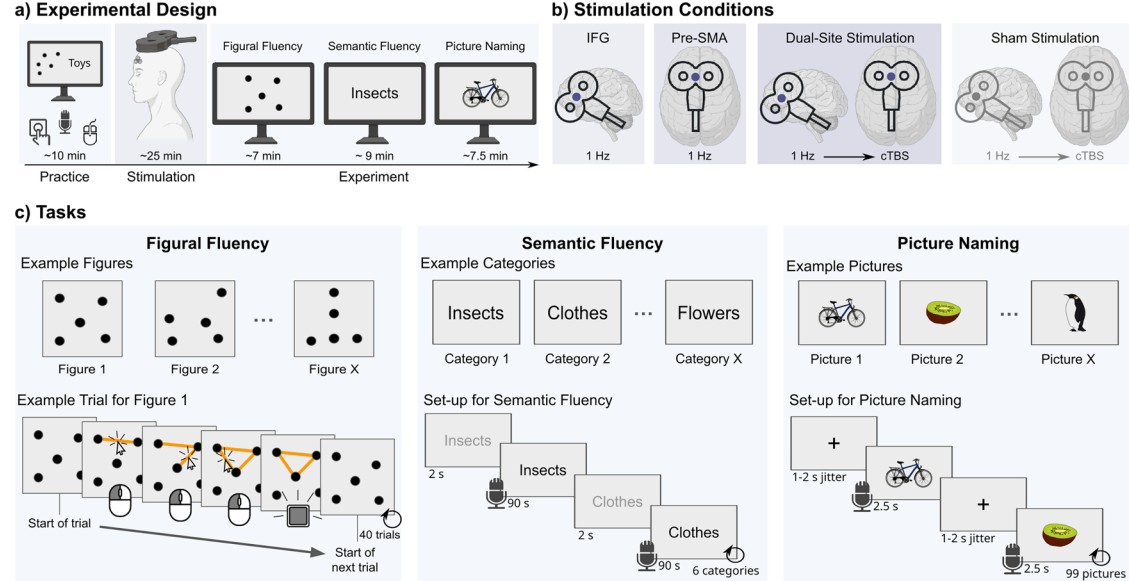

**Fig. 1 | Overview of the study design. a** At the beginning of each session, participants (re-)familiarized themselves with the different tasks. They then received "offline" rTMS and subsequently performed the experiment. The order of figural and semantic fluency in the task battery was counterbalanced across participants. **b** All participants received once single-site 1 Hz rTMS to the left IFG and pre-SMA, dual-site rTMS with 1 Hz to the left IFG, succeeded by cTBS to the pre-SMA, and sham rTMS replicating the dual-site condition with a placebo coil. **c** During each session, participants completed three blocks of the figural fluency task, six categories of semantic fluency, and one run (99 stimuli) of picture naming. Parts of the figure were created using BioRender. Martin, S. (2025) https://BioRender.com/d51w218.

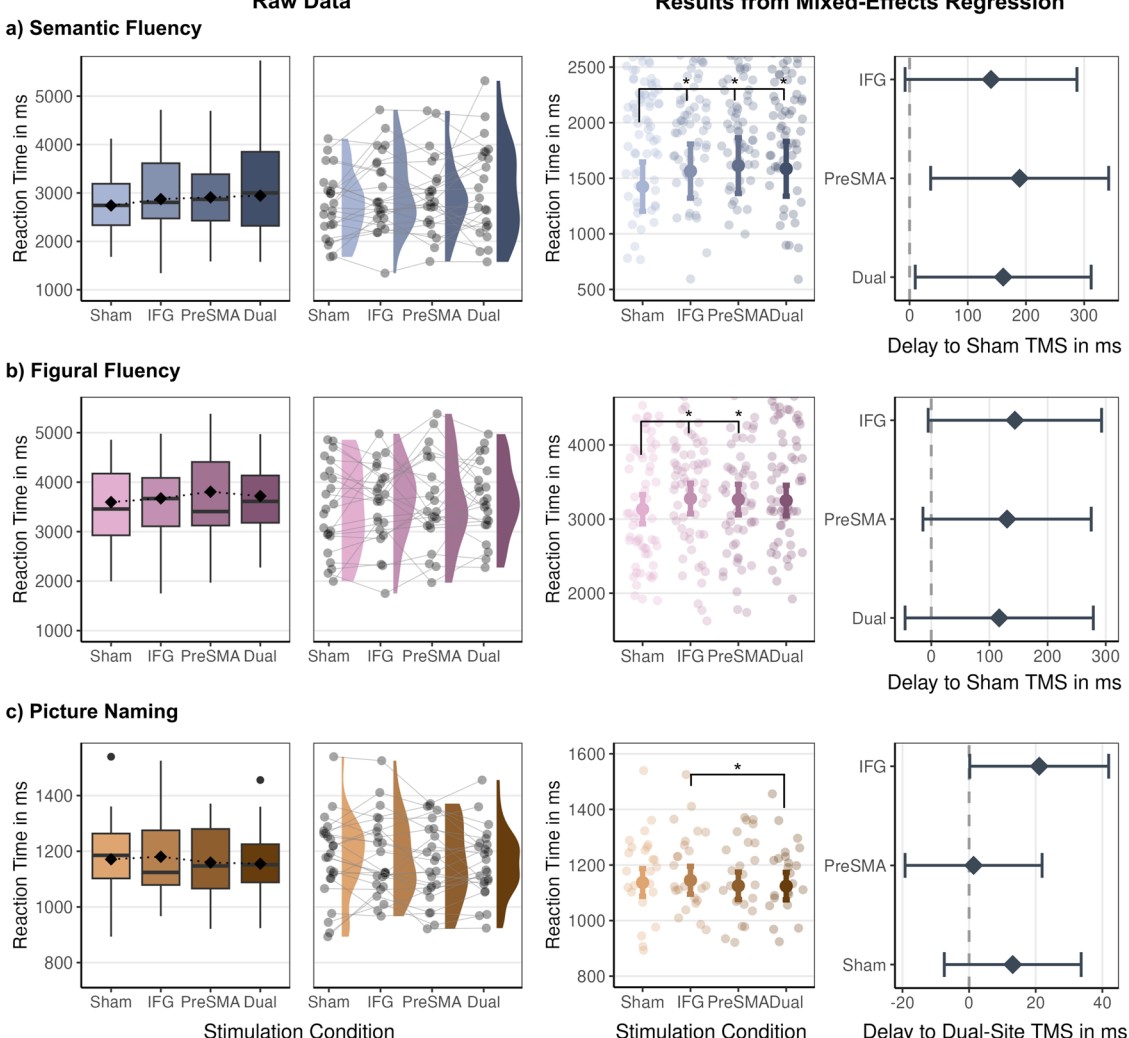

**Fig. 2 | Reaction times: raw data and regression results for each task.** Plots on the left side of panels **a**, **b**, and **c** show data aggregated by stimulation condition (boxplots) and distribution of raw data (half-violin plots) with means per participant for each stimulation condition as points. Plots on the right side panels **a**, **b**, and **c** show estimated marginal means from mixed-effects regression models and forest plots displaying the TMS-induced delay in reaction time. Results marked as *$p < 0.05$ are significant after FDR correction. $N = 24$ participants. In each boxplot, the center line indicates the median, the box edges represent the first and third quartiles, the whiskers extend to the minima and maxima, and the square shows the mean. Half-violin plots display the distribution of raw data. Error bars display the upper and lower bounds of 95% confidence intervals around the predicted value.

(*Incidence rate ratio*$(IRR)_{\text{IFG}} = 0.97$, $p = 0.506$, $IRR_{\text{Pre-SMA}} = 0.96$, $p = 0.506$, $IRR_{\text{Dual site}} = 0.94$, $p = 0.330$, Fig. 3a and Supplementary Table S3). However, results showed a significant effect of session, such that participants produced more correct items during session 4 compared to session 1 ($IRR_{\text{Session 4}} = 1.10$, $p = 0.006$, Supplementary Fig. S3a). Moreover, there were fewer correct items for the difficult categories than the easy categories ($IRR_{\text{Difficult categories}} = 0.61$, $p < 0.001$, Supplementary Fig. S3a) and higher accuracy with better executive abilities ($IRR_{\text{Executive abilities}} = 1.15$, $p = 0.003$, Supplementary Fig. S3a).

For figural fluency, rTMS to both the left IFG and pre-SMA significantly slowed reactions ($\beta_{\text{IFG}} = 0.04$, $p = 0.046$, $\beta_{\text{Pre-SMA}} = 0.04$, $p = 0.046$, Fig. 2b, Supplementary Table S2). The largest effect on RTs in figural fluency was observed for rTMS to the left IFG, with reactions being 144 ms slower compared to sham stimulation. Moreover, results showed a main effect of session, with faster reactions across sessions ($\beta_{\text{Session 2}} = -0.08$, $\beta_{\text{Session 3}} = -0.14$, $\beta_{\text{Session 4}} = -0.15$, all $p < 0.001$, Supplementary Fig. S3b), indicating a learning effect in this task. Results also revealed significant effects of the number of bars clicked ($\beta_{\text{N of bars clicked}} = 0.24$, $p < 0.001$, Supplementary Fig. S3b), with slower

reactions when participants created more complex designs with more bars activated, and accuracy ($\beta_{\text{Accuracy}} = -0.32$, $p = 0.001$, Supplementary Fig. S3b), such that higher accuracy was linked to faster reactions. There was also a main effect of executive abilities ($\beta_{\text{Executive abilities}} = -0.08$, $p = 0.026$, Supplementary Fig. S3b). Similar to semantic fluency, better executive abilities were associated with faster reactions in the figural fluency task.

Analyzing accuracy in the figural fluency task showed worse performance after rTMS to pre-SMA and dual-site compared to sham stimulation ($Odds\ Ratio(OR)_{\text{IFG}} = 1.17$, $p = 0.239$, $OR_{\text{Pre-SMA}} = 0.96$, $p = 0.736$, $OR_{\text{Dual site}} = 0.79$, $p_{\text{uncorr}} = 0.047$, $p_{\text{FDR}} = 0.051$, Fig. 3c and Supplementary Table S3). Moreover, results showed that participants improved their performance in session two compared to session one ($OR_{\text{Session 2}} = 1.45$, $p = 0.003$, Supplementary Fig. S3b) and when they activated a larger number of bars in a figural design ($OR_{\text{N of bars clicked}} = 1.21$, $p < 0.001$, Supplementary Fig. S3b).

Analyzing RTs in picture naming showed no significant effect of effective relative to sham rTMS ($\beta_{\text{IFG}} = 0.01$, $p = 0.381$, $\beta_{\text{Pre-SMA}} = -0.01$, $p = 0.203$, $\beta_{\text{Dual site}} = -0.01$, $p = 0.185$, Fig. 2c and Supplementary Table S2). However, post-hoc tests revealed significantly slower reactions after single-

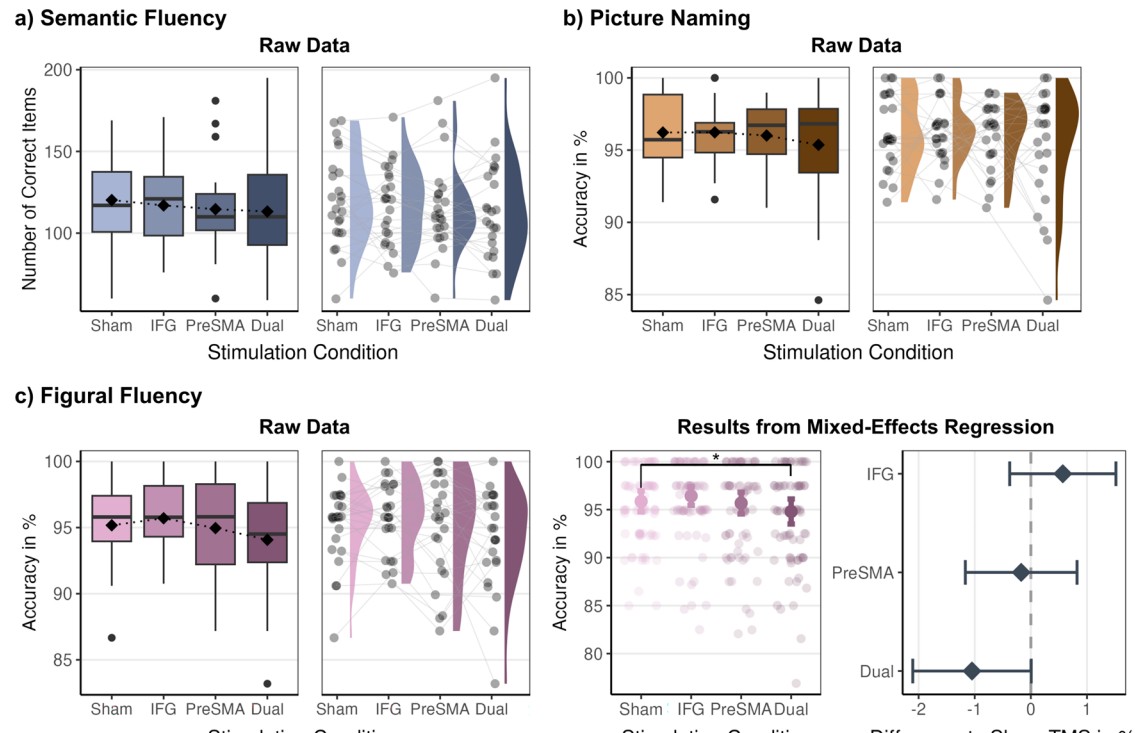

**Fig. 3 | Accuracy: raw data and regression results for each task.** Panels **a**, **b**, and the left side of **c** show data aggregated by stimulation condition (box plots) and distribution of raw data (half-violin plots) with means per participant for each stimulation condition as points. Plots on the right side of panel **c** show estimated marginal means from logistic regression for accuracy in figural fluency and a forest plot displaying the TMS-induced performance decrease. Results marked as *$p \le 0.05$ are significant after FDR correction. $N = 24$ participants. In each boxplot, the center line indicates the median, the box edges represent the first and third quartiles, the whiskers extend to the minima and maxima, and the square shows the mean. Half-violin plots display the distribution of raw data. Error bars display the upper and lower bounds of 95% confidence intervals around the predicted value.

site stimulation to left IFG compared to dual-site stimulation ($\beta_{\text{IFG-Dual-site rTMS}} = 0.02$, $p = 0.046$), with an average latency of 21 ms. Additionally, we observed a main effect of trial index, with reaction times becoming slower as the task progressed (trial index = 0.0003, $p = 0.003$; Supplementary Fig. S3c). There was also a main effect of executive abilities, such that higher executive abilities were associated with faster reaction times (executive abilities = −0.06, $p = 0.032$; Supplementary Fig. S3c).

Similarly, effective rTMS did not impact accuracy in picture naming ($OR_{\text{IFG}} = 1.00$, $p = 0.989$, $OR_{\text{Pre-SMA}} = 0.95$, $p = 0.905$, $\beta_{\text{Dual site}} = 0.81$, $p = 0.374$, Fig. 3b and Supplementary Table S3). Results revealed higher accuracy in all sessions relative to session one ($OR_{\text{Session 2}} = 2.09$, $OR_{\text{Session 3}} = 1.36$, $OR_{\text{Session 4}} = 2.09$, all $p < 0.05$, Supplementary Fig. S3c), as well as with progressing time within a session ($OR_{\text{Trial index}} = 1.01$, $p = 0.008$, Supplementary Fig. S3c), and with higher scores in the Spot-the-Word test[25] (STW), which we used as a measure of verbal intelligence ($OR_{\text{STW}} = 1.20$, $p = 0.035$, Supplementary Fig. S3c).

### Task-specific disruption of fluency by stimulation of left IFG and Pre-SMA

To test for a relationship between the induced stimulation strength over the target areas and a change in performance, we correlated individual e-field intensities (Fig. 4a) with mean RTs and accuracy of the respective task and stimulation session. Results showed a significant increase in RTs with stronger e-fields over the left IFG for semantic fluency ($r = 0.42$, $p_{\text{FDR}} = 0.039$, Fig. 4b). Moreover, linear regression combining both e-fields for sessions with dual-site stimulation showed poorer accuracy in figural fluency with increasing intensity of the e-field in pre-SMA ($r = -0.56$, $p = 0.043$, Fig. 4b). Supplementary Fig. S4 shows results for all correlations.

We were interested in the specific strength of the TMS effect in sub-regions of each stimulated cortical area and extracted e-field values in

cytoarchitectonically defined ROIs. Comparing e-field strengths between ROIs and stimulation sessions revealed a significant interaction between both predictors ($X = 1648.44$, $p < 0.001$), with higher e-field values in IFG and pre-SMA ROIs in the respective stimulation session (Fig. 4c, Supplementary Table S4). Furthermore, in the IFG session, the anterior IFG (BA45) received significantly more stimulation than the posterior IFG (BA44; $\beta_{\text{BA45-BA44}} = 8.87$, $p < 0.001$) and during rTMS over pre-SMA, the left pre-SMA showed stronger stimulation strength than the right pre-SMA ($\beta_{\text{left pre-SMA-right pre-SMA}} = 5.47$, $p = 0.004$).

Finally, we defined an off-target region in the posterior middle frontal gyrus (MFG) to further assess the spatial specificity of the TMS effects targeting the IFG. E-field modeling confirmed that the induced stimulation strength in the posterior MFG was significantly lower than in the targeted IFG ($\beta_{\text{BA45-MFG}} = 14.88$, $p < 0.001$) and pre-SMA regions ($\beta_{\text{left pre-SMA-MFG}} = 20.59$, $p < 0.001$, Fig. 4c) during the respective session, supporting the focality of our stimulation protocol. Moreover, unlike the IFG and pre-SMA, the e-field in the posterior MFG showed no relationship with behavioral performance, reinforcing the conclusion that observed behavioral effects are specific to stimulation of the intended target areas (Fig. 4d).

### TMS significantly impairs strategic processes in semantic fluency

TMS may induce behavioral changes extending beyond accuracy and reaction times, potentially altering task strategies. We investigated whether rTMS influenced clustering and switching, two core processes for cognitive flexibility in semantic fluency, by comparing the number of switches between stimulation conditions. Results showed a significant effect of pre-SMA and dual-site rTMS but not IFG rTMS ($IRR_{\text{Pre-SMA}} = 0.90$, $p = 0.039$, $IRR_{\text{Dual site}} = 0.87$, $p = 0.007$, $IRR_{\text{IFG}} = 0.93$, $p = 0.143$, Fig. 5 and

**Fig. 4 | Task-specific performance disruptions induced by the electric field (e-field) strength.** **a** Induced e-fields over our stimulation targets in the left IFG and pre-SMA. **b** Significant correlations of e-fields with behavior after FDR correction. Stronger e-fields in the IFG were associated with slower reactions during semantic fluency, while stronger e-fields in the pre-SMA were linked to poorer performance in figural fluency. **c** E-field strength in subregions of our stimulation targets, anterior (BA45) and posterior (BA44) IFG, and left and right pre-SMA, and in an adjacent off-target region, the left posterior middle frontal gyrus (MFG). Anatomical parcels are based on cytoarchitectonic probabilistic maps (Julich-Brain Atlas, Human Brainnetome Atlas). E-field strength was significantly stronger in our target site in IFG (BA45) and left pre-SMA compared with neighboring regions. **d** Correlations between e-fields and behavioral measures. For reaction times (RT) in semantic fluency, only the e-field in the IFG shows an effect. For accuracy in figural fluency during dual-site stimulation, multiple regression including both the IFG and pre-SMA e-fields reveals a significant effect for the pre-SMA, whereas regression with the MFG and pre-SMA e-fields does not. $N = 24$ participants. Ribbons of line graphs show 95% confidence intervals. Bars indicate group means ± standard error.

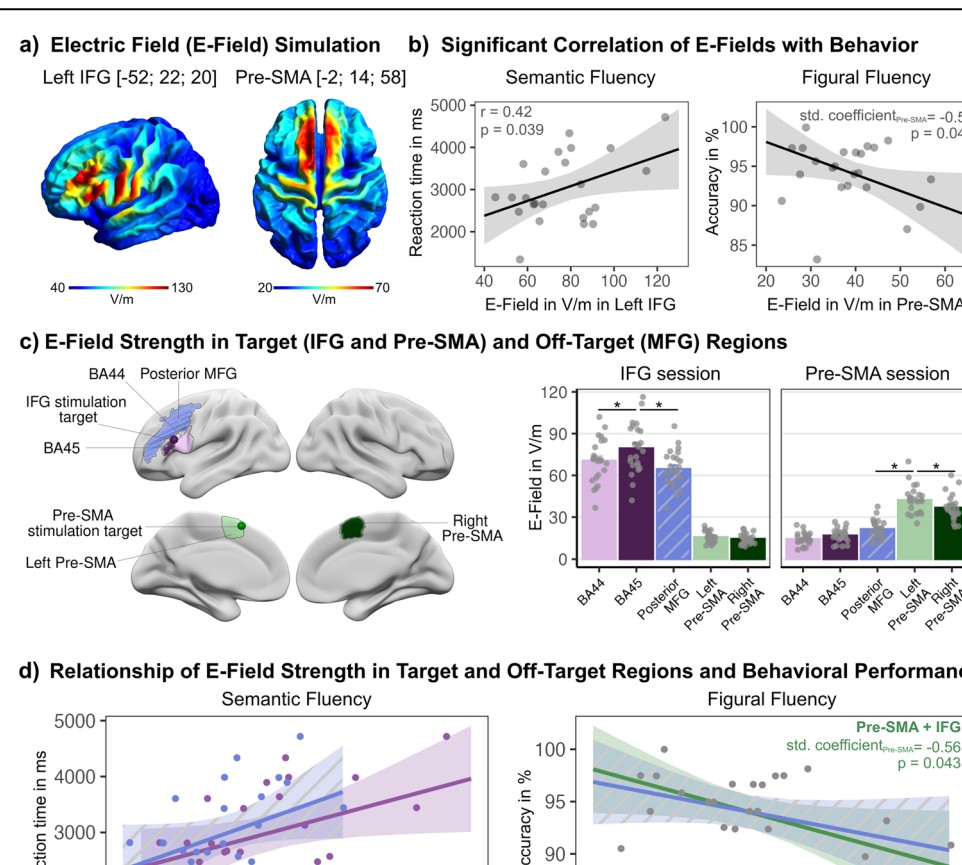

Supplementary Table S5). Both pre-SMA and dual-site rTMS reduced the number of switches compared to sham stimulation. The most pronounced effect was observed with dual-site rTMS, which resulted in an average reduction of 1.25 switches compared to sham stimulation. Moreover, there was an effect of category difficulty with less switches for difficult than easy categories ($IRR_{Difficult\ categories} = 0.60$, $p < 0.001$), but no effect of session (all $p > 0.05$, Supplementary Table S5). Supplementary Fig. S5 shows the average number of switches per semantic category.

## Discussion

This study provides new insight into the functional relevance of the left IFG and pre-SMA in semantic-specific and domain-general executive control. While stimulation of either region broadly disrupted both semantic and figural fluency, suggesting some shared functionality, we identified distinct specializations through the analysis of stimulation strength and behavioral outcomes. The left IFG was primarily linked to semantic control, as evidenced by its association with verbal fluency deficits, whereas the pre-SMA was more involved in domain-general executive control, impacting non-verbal fluency and cognitive flexibility, such as clustering and switching during semantic fluency. Additionally, dual-site TMS targeting both the IFG and pre-SMA delayed semantic fluency and impaired accuracy in figural fluency. Notably, the impaired accuracy in figural fluency was only observed after dual-site stimulation, providing fthe irst evidence for successful compensation of executive functions through either the left IFG or pre-SMA following perturbation of the other area. Overall, these findings stress the multi-dimensionality of cognitive control. They support a hierarchical model where semantic tasks recruit both specialized (IFG-mediated) and general executive (pre-SMA-mediated) resources, and demonstrate that executive control is based on a closely interacting network subserved by the left IFG and pre-SMA.

Contrary to our hypothesis, perturbation of the IFG not only impaired semantic but also figural fluency, indicating a functional role of this area in a non-verbal, strongly executive task. This finding aligns with a growing body of research suggesting the involvement of the IFG in multiple cognitive domains beyond language and semantics[4,26,27], including social cognition[28] and executive control[10,29]. However, what remains unclear is whether these different functions are reflected by distinct subregions within this cortical area or whether they show at least some functional overlap. Our target coordinate in the IFG was located in the dorsal anterior IFG (corresponding to Brodmann area 45), which has been repeatedly associated with semantic processing based on functional activity and connectivity[13,27,30,31]. The observed effect on figural fluency might be due to several reasons. First, recent investigations revealed a functional organization of the IFG along gradient axes where the dorsal IFG has been attributed a domain-general function of executive control[7,8], aligning with previous observations that the more language-specific anterior IFG is neighbored by a domain-general posterior region[10].

Second, our target coordinate was proximal to the posterior IFG (corresponding to Brodmann area 44). Although functional divisions do not strictly follow cytoarchitectonic borders, the stimulation likely affected this more domain-general region, influencing both fluency tasks. This aligns with van der Burght et al.[30], who demonstrated the challenge of dissociating TMS effects between anterior and posterior regions in the left IFG. Our e-field analysis of subregions revealed that while stimulation was statistically stronger in the anterior IFG, the posterior region still received substantial stimulation (71 V/m on average). To address this lack of focality, we calculated the effective induced stimulation strength, which is shaped by individual anatomical characteristics, at our target sites[22]. Crucially, this analysis unveiled a domain specificity for each region: only TMS over the IFG significantly impacted verbal fluency. This finding suggests that despite

**Fig. 5 | Stimulation-induced changes in the number of switches during semantic fluency. a** Raw data in switching for each stimulation condition. **b** Both pre-SMA and dual-site rTMS disrupt semantic fluency through fewer switches within semantic categories. Results marked as *$p < 0.05$ are significant after FDR correction. $N = 24$ participants. In each boxplot, the center line indicates the median, the box edges represent the first and third quartiles, the whiskers extend to the minima and maxima, and the square shows the mean. Error bars display the upper and lower bounds of 95% confidence intervals around the predicted value.

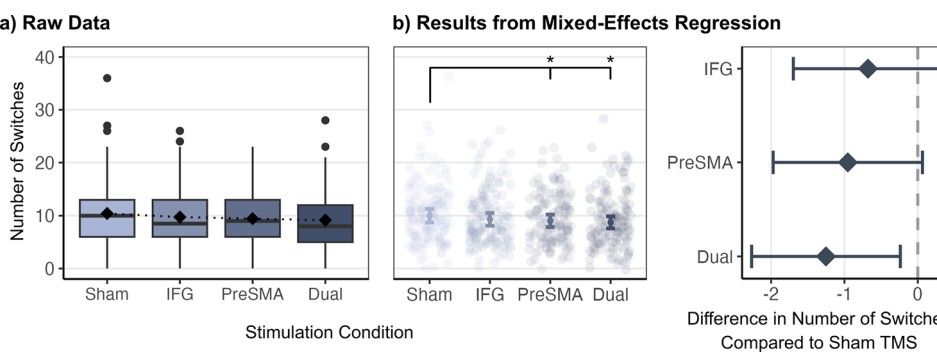

potential stimulation spread, the anterior IFG maintains a primary role in semantic control.

The effect of pre-SMA stimulation on both fluency tasks aligned with our expectations and confirms its domain-generality. The pre-SMA is considered a hub of the MDN[32,33], critical for response inhibition, cognitive flexibility, and conflict monitoring[34–36]. More recently, it has also been linked to semantic control processes in neuroimaging[5,7] and neurostimulation[37] research, as well as to compensatory roles in language processing in the aging brain[12,38] and post-stroke aphasia[39,40]. Importantly, our findings support an overall executive aspect of this region, which exhibits broad domain generality and involvement in executive control across domains. This conclusion is strengthened by the impaired accuracy in figural fluency with increasing pre-SMA stimulation and the reduced number of switches in semantic fluency following TMS to the pre-SMA. Switching between subcategories in semantic fluency requires cognitive flexibility and has been identified as an indicator of the successful search process during semantic fluency[41,42] and a detector of cognitive impairments[20,43]. Our results confirm the strongly executive nature of fluency tasks[14,16,44]. Moreover, they reconcile the role of the pre-SMA in semantic control with its overarching function in coordinating distributed executive resources during complex tasks.

We applied dual-site TMS, targeting the IFG followed by the pre-SMA, to test whether disrupting both areas would result in greater fluency deficits compared to single-site stimulation[45]. While no additional effect of dual-site TMS was observed for semantic fluency, figural fluency accuracy was impaired exclusively after dual-site stimulation. E-field analysis revealed that this effect was linked to the stimulation strength in the pre-SMA. This finding further underscores the key role of the IFG in domain-general executive control[46] and aligns with the delayed reactions in figural fluency after single-site TMS to both the IFG and pre-SMA. It is particularly noteworthy that we only observed this effect after dual-site TMS, suggesting that when either site is perturbed individually, successful compensatory mechanisms may mitigate the impact of TMS, at least for response accuracy. The fact that response efficiency was already affected after uni-site stimulation over either area converges with previous reports that response speed may be generally more sensitive to stimulation-induced perturbations than accuracy[45,47]. The significant impact of dual-site TMS on figural fluency accuracy highlights the importance of both regions in executive control, while the absence of this effect in semantic fluency further strengthens the unique capacity of the IFG in both domain-specific semantic and domain-general executive control.

Finally, we used picture naming as a low-level task which requires little cognitive control in healthy young adults but, like semantic fluency, engages targeted lexico-semantic retrieval. Consistent with this expectation, we observed delayed reactions in picture naming after TMS to the IFG, supporting its established role in naming processes[48–50]. No other stimulation-induced effects were observed for this task, likely due to participants' fast and highly accurate performance and the experimental design, where picture naming was always presented last, diminishing the temporary TMS effect.

In this study, we accounted for variation in executive control demands by including empirically defined "easy" and "difficult" categories in the semantic fluency task and by piloting and matching dot configurations for figural fluency. Although category difficulty robustly predicted performance, we did not observe a significant interaction between executive control demand and stimulation condition, indicating that TMS effects were not limited to the most demanding items. These findings suggest that TMS-induced disruptions affected overall task performance, supporting the involvement of the IFG and pre-SMA in domain-general executive processes. Finally, it should be noted that the anatomical definition of the left IFG in our study specifically excluded the pars orbitalis, as our stimulation site was located at the border of Brodmann areas 44 and 45. This limitation should be considered when interpreting the generalizability of our findings to the broader IFG. Moreover, the post-hoc e-field simulation in the posterior MFG showed similar strength and direction to the e-field in our target region IFG, though statistically with reduced strength and not the same relationship with behavioral performance. Future research is needed to further disentangle the contribution of IFG and posterior MFG to different domains of cognitive control.

In summary, this study probed the functional relevance of the left IFG and pre-SMA in semantic and executive control by inhibiting either site alone or together using rTMS. Examining the impact of these disruptions on verbal and non-verbal fluency tasks revealed significant contributions of both regions to domain-specific and domain-general control processes. The left IFG was found to be crucial for semantic control, particularly in verbal fluency tasks, while the pre-SMA played a more domain-general role in executive functions, affecting both semantic and non-semantic tasks. Notably, only dual-site stimulation impaired accuracy in figural fluency, suggesting compensatory mechanisms between these regions in executive control that help to maintain accurate responses even if response efficiency is reduced as evidenced by delayed response speed after uni-site stimulation. Our findings support a hierarchical model of cognitive control where semantic tasks engage both specialized and general executive resources. Furthermore, they provide evidence for a network of executive control subserved by the left IFG and pre-SMA, contributing to a deeper understanding of the neural basis of semantic and executive functions.

## Methods
### Participants
We tested 24 healthy adults ($M = 30.00$, SD = 5.32, range: 20–40 years, 13 female), who were right-handed, native German speakers, and had no contraindication to MRI and TMS. Vocabulary and verbal intelligence were assessed with the German version of the STW, and general executive functions with the TMT and the DSST. Participants were informed about experimental procedures but blinded to the different TMS conditions and their order. The study was approved by the local ethics committee of the Medical Faculty at Leipzig University and conducted in accordance with the Declaration of Helsinki. Written informed consent was obtained from each participant prior to the experiment. All ethical regulations relevant to human research participants were followed.

## Design

We employed a repeated-measures within-subjects design with four sessions per participant, which were separated by at least one week. At the beginning of each session, participants (re-)familiarized themselves with the task. They then received offline rTMS stimulation and subsequently performed the experiment (Fig. 1a). The study comprised four stimulation conditions: IFG, pre-SMA, dual TMS (IFG first, followed by pre-SMA stimulation), and sham stimulation (Fig. 1b). The order of these conditions was counterbalanced across subjects.

## Tasks and stimuli

Participants performed three tasks after offline rTMS: semantic fluency, figural fluency, and picture naming. The experiment always began with both fluency tasks to ensure maximal post-stimulation effects. Half of the participants started with semantic fluency, followed by figural fluency, and vice-versa for the others.

The semantic fluency task consisted of 24 semantic categories, containing easy (e.g., clothes) and difficult (e.g., insects) items (see Supplementary Fig. S5 for all categories). Difficulty ratings were taken from a previous study, for which we piloted difficulty levels of semantic categories[12]. Participants completed six categories during one session, consisting of three easy and three difficult categories, pseudorandomized in their order. For each category, participants were instructed to name as many unique items as possible during 90 s, avoiding repetitions and proper names (Fig. 1c).

We employed the Five-Points Test[51] (5PT), a standardized assessment of figural fluency[52], as a non-verbal measure of fluid intelligence. This test requires participants to generate abstract geometrical figures within a time limit, imposing executive demands similar to the semantic fluency task. We adopted a computer-based version of the 5PT, which we piloted online in a separate sample of 30 young adults ($M = 23.5$, $SD = 2.4$, range: 18–30 years). To this end, we developed eleven additional 5-dot arrangements as stimuli (Supplementary Fig. S1). Although the dot arrangements differed in geometrical regularity and internal symmetry to the original design of the 5PT, we controlled for dimensions on screen and ease of clickability by matching the cumulative bar length of each figure. Participants completed 3 different dot arrangements per session, each lasting a maximum of 3.5 min or until 40 designs were completed, and were instructed to produce as many unique designs as possible by connecting two or more dots with straight lines (Fig. 1c).

The picture-naming task was implemented as a low-level control task, accounting for bottom-up access processes implicated in word retrieval. We compiled four stimulus lists with 99 pictures each. Pictures were taken from the Multipic database[53] and balanced for naming agreement, visual complexity, word frequency, and word length in syllables (see Supplementary Table S1 and Supplementary Fig. S2 for parameters). Each picture was presented for 2.5 s, with a total duration of 7.5 min for picture naming per session (Fig. 1c). All tasks were programmed and presented with Psychopy version 2021.2.3.

## Transcranial magnetic stimulation

Before the experimental tasks, we applied 1 Hz offline rTMS at 100% resting motor threshold (rMT) for 25 min over either the left IFG or pre-SMA. During the session with dual-site stimulation, 1 Hz rTMS to the left IFG was immediately succeeded by continuous theta burst stimulation (cTBS) to pre-SMA. cTBS was delivered at 90% rMT[cf 54]. and consisted of 600 pulses delivered in 50 Hz triplets every 200 ms for a total of 40 s. We chose to employ cTBS as a second stimulation protocol during dual-site TMS to avoid extended stimulation time during this session. To maximize effective blinding, the sham session mimicked the dual-site stimulation using a placebo coil.

The rMT was assessed at the beginning of the first session and defined as the lowest stimulation intensity inducing motor evoked potentials of $\geq 50$ µV at least 5/10 times in the relaxed first dorsal interosseous muscle when single-pulse TMS was applied to the left motor cortex. TMS was delivered with a figure-of-eight coil (MagVenture MCF-B65 for effective stimulation and MCF-P-B65 placebo coil for sham stimulation) connected to a MagPro

X100 stimulator (MagVenture) and guided by stereotactic neuronavigation (TMS Navigator, Localite). To this end, the participant's head was co-registered onto an individual structural MR scan taken from the in-house database or newly acquired at a 3 T Siemens MR scanner. Anatomical data were preprocessed using SPM12 (Wellcome Trust Center for Neuroimaging) in Matlab (version R2022b). The coil was oriented at 0° for pre-SMA[55] and at 45° for IFG stimulation (Fig. 1b). Target locations for the left IFG ($x$: −52, $y$: 22, $z$: 20) and pre-SMA ($x$: −2, $y$: 14, $z$: 58) were derived from a meta-analysis on activation peaks in semantic fluency tasks[31]. The MNI coordinates were converted into participants' native space and used for neuronavigation.

We performed post-hoc e-field simulations using SimNIBS v.4.0.0[56] to characterize the location, extent, and strength of the e-field induced by rTMS over the pre-SMA and IFG in each participant. E-field values were extracted by defining a region of interest with a 5 mm radius around the stimulation coordinates. From this region, values at the 95% percentile were extracted from the individually calculated e-fields, focusing exclusively on gray matter. To assess the relationship between the induced stimulation strength over the target areas and a change in performance, we correlated the individual by-target-site induced e-field with the individual mean reaction time (RT) and accuracy per task and session. For the session with dual-site stimulation, we calculated linear regression models using individual e-fields from both targets as predictors and behavioral performance as outcome measure. To facilitate more meaningful comparisons between model coefficients and correlation values, model parameters were standardized after the analysis. Standardization places coefficients on a common scale, allowing for direct comparison of effect sizes across predictors and with correlation coefficients.

To further understand how the stimulation affected subregions in the IFG and pre-SMA, we additionally extracted e-field values from cytoarchitectonic probabilistic regions of interest (ROIs) in both cortical areas, based on the Julich-Brain Atlas[57]. Specifically, we included anterior (BA45) and posterior IFG (BA44), as well as left and right pre-SMA. All ROIs were thresholded at 35%, binarized, and resampled to subject space before extracting the 95th percentile (95% quantile) of the e-field distribution within each parcel. A linear mixed-effects model with ROI and session (stimulation target: IFG or pre-SMA) as fixed effects, and random intercepts for participants as well as by-participant random slopes for session, was used to compare e-field values across target ROIs.

Furthermore, to evaluate off-target stimulation effects, we also included the posterior middle frontal gyrus (MFG) as a comparison region, using a probabilistic parcel from the Human Brainnetome Atlas (Fan et al., 2016)[58]. We compared the relationship between the e-field strength in our IFG target and the MFG off-target to explore the spatial specificity of TMS.

## Data analysis

**Preprocessing.** Recordings from the semantic fluency and the picture naming tasks were transcribed by three native German raters who annotated reaction times with Praat[59] and attributed accuracy scores to each response. For semantic fluency, repetitions and wrong answers were marked as incorrect. For picture naming, answers that did not match with the names prescribed by Multipic's norms were evaluated by the raters. For figural fluency, repeated designs were marked as incorrect.

We obtained RTs for each task according to the following criteria: For semantic fluency, we extracted the interval between the offset of the previous item and the onset of the following item. For figural fluency, RTs were calculated as the interval between the presentation of a blank set of dots and the subject's button press to record their design, resulting in a maximum of 40 trials per figure. For picture naming, we considered the interval between picture presentation and naming onset. To exclude excessive outlier values from RTs, we employed a lenient approach where, for each participant, session, task, and stimulus item (category, figure), RTs at least three SDs above the mean were excluded. This procedure removed 3.6% of all trials.

**Statistical analysis.** For each task, accuracy and RTs of each session were analyzed using mixed-effects models. Linear regression models were

**Table 1 | Mixed-effects models for reaction times and accuracy**

| Task | DV | Fixed Effects | Random Effects |
|---|---|---|---|
| Semantic Fluency | log(RT) ~ | Stimulation condition + Session + Category difficulty + Task order + Trial index + Spot-the-Word test + Neuropsychological score + | (1 + Category difficulty \| Participant) + (1 \| Stimulus list) + (1 \| Response) |
| | N correct ~ | Stimulation condition + Session + Category difficulty + Spot-the-Word test + Neuropsychological score + | (1 \| Participant) + (1 \| Stimulus list) |
| Figural Fluency | log(RT) ~ | Stimulation condition + Session + Stimulus list + N Bars clicked + Accuracy + Task order + Trial index + Neuropsychological score + | (1 + Stimulus list \| Participant) + (1 \| Design) + (1 \| Response) |
| | Correct ~ | Stimulation condition + Session + N Bars clicked + Trial index + Neuropsychological score + | (1 \| Participant) + (1 \| Design) |
| Picture Naming | log(RT) ~ | Stimulation condition + Session + Accuracy + Task order + Trial index + Spot-the-Word test + Neuropsychological score + | (1 \| Participant) + (1 \| Picture) |
| | Correct ~ | Stimulation condition + Session + Trial index + Spot-the-Word test + Neuropsychological score + | (1 \| Participant) + (1 \| Picture) |

*DV* dependent variable, *RT* reaction time.

applied to log-transformed RTs, considering only correct trials. Accuracy data were analyzed using logistic regression for the binomially distributed data of figural fluency and picture naming, and negative binomial regression for the count data of semantic fluency.

Table 1 presents the task-specific models for reaction time and accuracy. Based on our research question, stimulation condition (i.e., sham, pre-SMA, IFG, or dual-site rTMS) was always included as a fixed effect. Additionally, all models incorporated fixed effects for session to account for potential learning effects and trial index to account for the linearly decreasing TMS effect during a session. Moreover, we included scores from neuropsychological tests as measures of verbal intelligence and executive abilities. Both executive tests (TMT, DSST) were combined into a single score. This was done by first inverting the reaction times of the TMT, then z-standardizing both the TMT and DSST, and finally calculating the individual mean of both tests. Scores from the Spot-the-Word test were z-standardized and included as a separate regressor, as it assesses verbal knowledge without time restrictions and thus represents a different cognitive concept. Finally, each model included task-specific predictors, such as category difficulty for semantic fluency and number of bars clicked for figural fluency.

Stepwise model selection was employed to determine the best-fitting model based on the Akaike Information Criterion (AIC) and model convergence, with a change of at least two points in AIC considered meaningful[60]. All models included at least two random intercepts to account for the variance introduced by participants and stimuli. Where possible, RT models also included task-specific by-participant random slopes. All factors were contrast-coded using the simple coding scheme, with *sham stimulation* and *session 1* as reference levels. As generalized linear mixed-effects models for accuracy were more prone to convergence issues and singularity warnings than linear mixed-effects models for reaction time, simpler model structures were chosen.

Statistical models were conducted using R v.4.4.1[61] with the lme4 package[62] for mixed-effects models and the performance package[63] for model comparisons. Post-hoc multiple comparisons were performed using the emmeans package[64] to determine effects between individual factor levels other than the reference levels, with FDR correction applied. Results with FDR-corrected *p*-values < 0.05 were considered significant. Plots were generated using ggplot2[65] and ggeffects[66], and model output via sjPlot[67]. All reported *p*-values for stimulation condition are FDR-corrected to account for comparisons of each condition to sham in the output of the mixed-effects models.

**Exploratory quantitative analysis for semantic fluency.** To further investigate the effects of TMS on the executive components of verbal fluency, we conducted an exploratory quantitative analysis on clustering and switching strategies within the verbal fluency task. To evaluate participants' lexical retrieval, we employed FastText[68] for fully automated switch detection. This method estimates the number of switches among clusters by calculating the cosine similarity between the vector of any uttered word and the subsequent word vector. If the cosine similarity falls below a predefined switch threshold, a new cluster is identified. We defined category-specific thresholds as the median similarity across all word pairs within a given semantic category, as suggested by Alacam et al.[69]. We thus counted a cluster as two or more consecutive words above the predefined switch threshold and a switch each time a participant transitioned from one cluster to a different subcategory or produced a word that did not belong to the previous cluster. While clustering is thought to reflect the integrity of semantic memory and temporal lobe processes[42], switching indexes executive control abilities, specifically, cognitive flexibility and strategic search, which are supported by frontal lobe function[42] and were thus hypothesized to be affected by TMS. To test this hypothesis, we ran a negative binomial mixed-effects regression with the number of switches per participant and category as the dependent variable and stimulation condition, session, and category difficulty as fixed effects. Moreover, we included random intercepts for participants and categories.

## Reporting summary
Further information on research design is available in the Nature Portfolio Reporting Summary linked to this article.

## Data availability
Data are available in our OSF repository[70].

## Code availability
Analysis scripts are available in our OSF repository[70].

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

## Acknowledgements
We would like to thank Julia Siodmiak for assistance with data acquisition. Moreover, we thank all student assistants and interns involved in the transcriptions of verbal fluency and picture naming data. We thank Ole Numssen for advice on the calculations of the electric fields. G.H. was supported by Lise Meitner Excellence funding from the Max Planck Society, the European Research Council (ERC-2021-COG 101043747), and the German Research Foundation (HA 6314/3-1, HA 6314/4-2, HA 6314/9-1). S.M. and M.F. were supported by the German Scholarship Foundation.

## Author contributions
S.M.: conceptualization, data curation, formal analysis, investigation, methodology, project administration, visualization, writing: original draft, review & editing. M.F.: conceptualization, data curation, formal analysis, investigation, methodology, project administration. A.B.: formal analysis, methodology, writing: review & editing. G.H.: conceptualization, funding acquisition, supervision, methodology, writing: review & editing.

## Funding

## Competing interests
The authors declare no competing interests.

## Additional information

**Peer review information** *Communications Biology* thanks Susan Teubner-Rhodes who co-reviewed with Anna Pusser, and Le Li for their peer review of this work. Primary handling editors: Christian Beste and Jasmine Pan. A peer review file is available.

