## [Transparent Peer Review file · Communications Biology]

Causal Contributions of Left Inferior and Medial Frontal Cortex to Semantic and Executive Control

Corresponding Author: Dr Sandra Martin

Version 0:

Reviewer comments:

Reviewer #1

(Remarks to the Author)

Martin and colleagues present a TMS study on semantic and executive control. They applied repetitive TMS over the left IFG, pre-SMA and both regions and sham stimulation across four sessions in 24 adults. After each stimulation, the participants performed a semantic fluency task, a non-verbal figural fluency task and a picture naming task. Mixed-effects regression showed that TMS over the left IFG, pre-SMA and both regions all induced increase in RT in both the semantic and the non-verbal fluency tasks. Dual-site TMS further decreased response accuracy in the non-verbal fluency task. The performance in either task across participants was correlated with e-field strength. Finally, TMS over the pre-SMA and both regions decreased the number of switches during the semantic fluency task. The authors conclude that the left IFG was primarily associated with semantic control, affecting semantic fluency, while the pre-SMA played a domain-general role in executive functions, affecting non-verbal fluency and cognitive flexibility. Overall, the study is interesting and address the causal contribution of brain regions to semantic and executive control. The experimental design and analyses are generally appropriate. However, I have several major and minor comments that should be addressed before I can recommend publication:

Major points:

1 Some statements appear inconsistent with the reported results, and this creates ambiguity.

For example, in the Abstract the author says that the results suggest “a flexible task-dependent contribution of the IFG”. However, for the key measure RT, TMS over the IFG impair performance in both the semantic fluency task and the non-verbal fluency task (Figure2a, 2b), indicating a task-general effect. I recommend revising all such statements that seem to contradict to the results.

2 The author used FDR to control for multiple comparisons. However, the p threshold derived from the FDR method is highly dependent on the specific set of p values being corrected. To increase the transparency of the results, please clarify original p values and the p thresholds derived from the FDR.

Minor points:

3 Figure4b, it is strange the p value for $r = -0.56$ is larger than that for $r = 0.42$? Please use the same criterion for reporting statistics.

4 It is unclear why the mixed-effects models are different between RT and accuracy. The authors could explain this.

5 To increase readability, I suggest to add brief introduction or definition of the following terms in the Results section when they were first mentioned: a main effect of session, category difficulty, executive abilities, etc.

6 Given TMS's limited spatial focality (as also claimed by the authors), could stimulation targeting the left IFG also affect the adjacent posterior MFG?

7 The left IFG in this study does not contain its orbital part, and the manuscript have not mentioned this point.

Reviewer #2

(Remarks to the Author)

Summary: This study used transcranial magnetic stimulation (TMS) to examine how two brain regions—the left inferior frontal gyrus (IFG) and pre-supplementary motor area (pre-SMA)—contribute to semantic control (retrieving meaning from memory) and executive control (general thinking skills like flexibility and inhibition). Researchers stimulated these areas in 24 healthy adults and had them complete verbal and non-verbal fluency tasks. They found that the left IFG mainly supports semantic control while pre-SMA is involved in domain-general executive control; however, since the accuracy of figural fluency was only affected after both regions were stimulated, this suggests that the regions can compensate for each other when enacting non-verbal executive control. The paper is well-written and makes an important contribution to the literature on brain regions that support semantic fluency as well as the domain-specificity versus -generality of these processes. Our major concerns relate to better delineating how the tasks measure executive control, given the absence of conditions that manipulate executive control demands, as well as better defining clustering and switching responses that are used to measure executive control processes. Describing these elements will be essential for conveying the theoretical impact of the work. We also have some clarification questions regarding the methods and analyses.

Major Comments:

1) While semantic fluency may broadly involve executive control, there are also specific conditions that place greater demands on executive control than others. For example, Novick et al. (2009) found that a patient with focal damage to the left IFG was selectively impaired in fluency for larger semantic categories (compared to smaller semantic categories), as the larger categories have greater selection demands due to there being more available correct responses. Similarly, picture naming engages left IFG to a greater extent when a picture is associated with more than one name (Kan & Thompson-Schill, 2004), suggesting greater executive control demands for pictures with low name agreement. While the authors employed tasks thought to involve executive control, they did not explicitly manipulate executive control demands in these tasks, limiting their ability to determine if TMS selectively disrupted conditions hypothesized to rely on executive control. We would like to see more discussion of the executive control demands of each task they used. In the absence of conditions manipulating executive control, how do we know that the TMS-induced disruptions are due to executive/semantic control per se as opposed to a different ability like processing speed that is also shared across semantic and figural fluency tasks?

Kan, I. P., & Thompson-Schill, S. L. (2004). Effect of name agreement on prefrontal activity during overt and covert picture naming. *Cognitive, Affective, and Behavioral Neuroscience*, 4, 43-57.

Novick, J. M., Kan, I. P., Trueswell, J., & Thompson-Schill, S. (2009). A case for conflict across multiple domains: Memory and language impairments following damage to ventrolateral prefrontal cortex. *Cognitive Neuropsychology*, 26(6), 527-567.

2) The authors examined clustering and switching of responses during verbal fluency to understand executive control effects. Given that executive control demands were not manipulated through task conditions, this analysis appears to be central to the research question of how TMS to left IFG/pre-SMA affects executive control during fluency. To this end, we would like to see more detail about what constituted clustering versus switching. How are these defined, both conceptually and operationally? How does clustering versus switching index executive control processes?

Minor Comments:

-In 64-67 - While the paper notes that participants completed four sessions with different stimulation conditions (IFG, pre-SMA, dual-site, sham), it is not immediately clear that each stimulation condition occurred in a separate session and that the dual-site stimulation (IFG followed by pre-SMA) was administered within a single session. Making this distinction more explicit here or in the figures would improve clarity.

-In 99 – “TMS over the left IFG and Pre-SMA both delay Semantic and Figure Fluency” -> It seems like “both” is in the wrong place. Consider either “TMS over both the left IFG and Pre-SMA” or “delay both Semantic and Figure Fluency.”

-In 136-137 & Figure 2C – Why is the TMS-related delay being compared to dual-site rather than sham stimulation for picture naming?

-In 138-140 – “...there was a main effect for trial index with slower reactions with progressing time... and for executive abilities..., which reduced reaction times” -> I initially misinterpreted this sentence to mean that slower reactions occurred for executive abilities (in addition to progressing time), until I got to the clause “which reduced reaction times,” indicating that executive abilities were associated with faster reactions. Consider rephrasing.

-In 172-174 – “poorer performance in figural fluency with increasing intensity of the e-field in pre-SMA ($r = -0.55$, $p = 0.043$, Fig. 4b).” -> Since the preceding sentence was talking about RTs in semantic fluency, it was not immediately clear that “poorer performance” here is referring to accuracy (i.e., slower response times might also be interpreted as poorer performance). Also, the r -value printed on Fig. 4b is -0.56 as opposed to -0.55 stated in the text.

-In 300-308 – Was a power analysis conducted to determine the sample size? The authors state that this is a “relatively large sample” given “recent methodological considerations for TMS studies in cognition,” but it is unclear what these

methodological considerations are or what their statistical power for effects of interest is.

-In 342 – Picture naming lists were balanced for several factors. As items with low name agreement and/or low frequency may increase executive control demands on this “low-level control task,” it would be useful to report some descriptive stats (M, SD or range) about these factors to demonstrate that picture naming was indeed a “low-level control task.”

-In 364-365 – What software/tools were used to convert MNI coordinates into native space?

-In 385-386 – What was the inter-rater reliability for RTs that were identified using Praat?

-In 400-401 – “Linear regression models were applied to log-transformed RTs” -> Why transform RTs to meet assumptions of linear regression rather than using GLMMs on the raw RT data (as recommended by Lo & Andrews, 2015)?

Lo S and Andrews S (2015) To transform or not to transform: using generalized linear mixed models to analyse reaction time data. *Front. Psychol.* 6:1171. doi: 10.3389/fpsyg.2015.01171

-In 401-403 – “Accuracy data were analyzed using logistic regression for the binomially distributed data of figural fluency and picture naming, and negative binomial regression for the count data of semantic fluency.” -> Why isn't figural fluency measured as count data as well? As a fluency task, it seems like the relevant metric is the number of unique figures produced in the allowed time.

-In 419-420 – “Where possible RT models also included task-specific by-participant random slopes.” -> What does ‘where possible’ mean? Is this determined by the design, by model convergence, or some other factor? Also, why were random slopes not considered for accuracy data (Table 1) or ROI e-field values (In 381-382). Not including random slopes can bias estimates (Barr et al., 2013).

Barr, D. J., Levy, R., Scheepers, C., & Tily, H. J. (2013). Random effects structure for confirmatory hypothesis testing: Keep it maximal. *Journal of Memory and Language*, 68(3), 255–278. <https://doi.org/10.1016/j.jml.2012.11.001>

Figure S2a – Why do the condition means for RT by each trial type appear to be less than or equal to the minimum participant RTs?

Reviewer #3

(Remarks to the Author)

I co-reviewed this manuscript with one of the reviewers who provided the listed reports. This is part of the Communications Biology initiative to facilitate training in peer review and to provide appropriate recognition for Early Career Researchers who co-review manuscripts.

Version 1:

Reviewer comments:

Reviewer #1

(Remarks to the Author)

The authors have thoroughly revised the data analysis, results, discussion and conclusion of the article. I have further suggestions as follow:

1. The newly added analyses of E-Field strength in left posterior MFG reported in Figure 4 are interesting. However, the effect size of relationship of E-Field strength and behavioral performance was comparable between the left posterior MFG and the left IFG ($r = 0.38$ vs. 0.42 ; and the positive correlation was marginally significant for the left posterior MFG). I suggest that the authors provide additional evidence demonstrating that TMS selectively disrupts left IFG function while leave MFG function intact.

2. Table S2. It is counterintuitive that TMS over either IFG or PreSMA individually affected reaction times in the figural fluency task, whereas concurrent TMS over both regions did not. I suggest adding possible explanations for this result.

Reviewer #2

(Remarks to the Author)

Summary: The authors addressed many of the previously raised concerns in their revised manuscript, which makes a valuable contribution to our understanding of the neural mechanisms supporting semantic and executive control. However, there are a few additional minor points that need to be clarified for the final publication.

Minor Comments:

p. 5 – The authors now state that executive ability was measured as the average of scores on TMT and DSST. On p. 17, the authors clarify that both metrics were converted to z-scores prior to averaging. This should be specified here as well to avoid confusion.

p. 9 – There is still a mismatch between the regression coefficient between e-field strength of the pre-SMA and accuracy in figural fluency reported in the text ($r = -0.55$) and Figure 4b ($r = -0.56$). The number was updated in the rebuttal letter but not in the main text.

p. 13 – “Although category difficulty robustly predicted performance, we did not observe a significant interaction between executive control demand and stimulation condition, indicating that TMS effects were not limited to the most demanding items.” -> As far as we can tell, the interaction between category difficulty and stimulation condition wasn’t tested: the models for semantic fluency in Table 1 indicate fixed effects of stimulation condition and category difficulty but not their interaction. The description of statistical analyses on p. 17-18 mentions that stepwise model selection was performed, but never mentions interactions being included as part of the model testing. If interactions were tested, this needs to be stated. Otherwise, it is not appropriate to conclude that TMS effects applied to all semantic fluency items, regardless of whether they were “easy” or “hard.”

p. 16 – The authors’ response clarifies that SPM12 was used to convert MNI coordinates into native space, but SPM12 is never cited in the text or reference list.

Throughout – We appreciate the clarification regarding clustering and switching during verbal fluency in the revised manuscript. Nevertheless, a clear definition of clustering and switching processes is not provided until the Methods on p. 18, well after the results of this analysis are reported (p. 9-10). I recommend defining these terms when they are first introduced on p. 3 of the Introduction. The authors could consider moving the clear descriptions that they provide in the Supplementary Materials (e.g., “Clustering refers to the production of words within a semantic or phonemic subcategory, such as listing several farm animals in a row when asked to name animals. Switching is defined as the ability to shift from one subcategory (cluster) to another, for example, moving from farm animals to zoo animals during the task”) to just after the sentence “Additionally, we employed a novel machine learning-based clustering and switching analysis to examine TMS effects on category switching during verbal fluency, a process highly sensitive to the detection of neurodegenerative diseases (Troyer et al., 1998)” on p. 3. Moreover, this would allow the removal of this section from the Supplementary Materials, as the rest of the information in that section is provided elsewhere in the text.

Table S1 & Figure S2 – The scales of measurement for word frequency and visual complexity need to be clarified.

Reviewer #3

(Remarks to the Author)

I co-reviewed this manuscript with one of the reviewers who provided the listed reports. This is part of the Communications Biology initiative to facilitate training in peer review and to provide appropriate recognition for Early Career Researchers who co-review manuscripts.

Version 2:

Reviewer comments:

Reviewer #1

(Remarks to the Author)

The authors have addressed all my concerns.

Reviewer #2

(Remarks to the Author)

The authors have addressed our remaining concerns. The revised manuscript makes a valuable contribution to our understanding of the neural mechanisms supporting semantic and executive control.

Reviewer #3

(Remarks to the Author)

I co-reviewed this manuscript with one of the reviewers who provided the listed reports. This is part of the Communications Biology initiative to facilitate training in peer review and to provide appropriate recognition for Early Career Researchers who co-review manuscripts.

Response Letter

We thank all reviewers for their positive feedback and constructive comments. We have addressed each point in detail below and believe that these revisions have substantially improved the quality and clarity of the manuscript, making it well-suited for publication.

Reviewer #1

Overall, the study is interesting and addresses the causal contribution of brain regions to semantic and executive control. The experimental design and analyses are generally appropriate. However, I have several major and minor comments that should be addressed before I can recommend publication:

Reply

Thank you for the positive evaluation of our work and your helpful comments.

Major Comments

Comment 1

Some statements appear inconsistent with the reported results, and this creates ambiguity. For example, in the Abstract the author says that the results suggest “a flexible task-dependent contribution of the IFG”. However, for the key measure RT, TMS over the IFG impair performance in both the semantic fluency task and the non-verbal fluency task (Figure 2a, 2b), indicating a task-general effect. I recommend revising all such statements that seem to contradict to the results.

Reply

We would like to apologize that the conclusions do not always seem consistent with the reported results. It is correct that TMS over the IFG impaired both semantic and figural fluency. However, we would like to emphasize that **after considering the individual field strength (electric field)** induced through TMS, we found a **task-specificity of IFG** stimulation only for reaction times of semantic fluency (Fig. 4b). We have revised the statement accordingly:

“These findings underscore the multidimensionality of cognitive control and suggest a flexible contribution of the IFG to control processes, either as semantic-specific or general executive resource.”

Comment 2

The author used FDR to control for multiple comparisons. However, the p threshold derived from the FDR method is highly dependent on the specific set of p values being corrected. To increase the transparency of the results, please clarify original p values and the p thresholds derived from the FDR.

Reply

Thank you for this constructive comment. We have now added the uncorrected p-values next to the p-values after FDR correction in the results tables (Supplementary Information). Please note that in the model output tables (Tables S1 and S2), FDR correction was applied exclusively to the p-values associated with the stimulation condition. This approach provides an additional level of statistical robustness by controlling for multiple comparisons within this specific set of tests. Both the original (unadjusted) p-values and the FDR-adjusted p-values are reported for full transparency. Moreover, the threshold value of each FDR correction is marked with an underscore. Finally, we added the following sentence to the Methods section (Statistical Analysis) on page 18:

“Results with FDR-corrected p-values < 0.05 were considered significant.”

Minor Comments

Comment 3

Figure 4b, it is strange the p value for $r = -0.56$ is larger than that for $r = 0.42$? Please use the same criterion for reporting statistics.

Reply

We apologize that we did not better clarify the origin of these values in our previous version. While the r - and p -value for the significant relation between the e-field strength in the IFG and reaction times for semantic fluency is based on a Pearson correlation, the statistical values for dual-site stimulation are based on multiple linear regression models with both e-fields as predictors and behavioral performance (accuracy in figural fluency in Figure 4b) as outcome variable. For better comparison, the parameter coefficient from the model output was standardized using the `standardize_parameters` function from the `parameters` package (Lüdtke et al., 2020). To clarify this point, we have now relabelled the value from the linear

regression model in Fig. 4b to “standardized coefficient”. Furthermore, we have added this information to the Methods section (Transcranial Magnetic Stimulation) on page 16:

“To facilitate more meaningful comparisons between model coefficients and correlation values, model parameters were standardized after the analysis. Standardization places coefficients on a common scale, allowing for direct comparison of effect sizes across predictors and with correlation coefficients.”

Comment 4

It is unclear why the mixed-effects models are different between RT and accuracy. The authors could explain this.

Reply

The mixed-effects models for RT and accuracy differ due to the nature of the outcome variables and statistical considerations. RT, as a continuous variable, was analyzed using linear mixed-effects models, while accuracy, being binary, required generalized linear mixed-effects models. Due to the increased complexity of GLMMs, these models are known to be more prone to convergence issues compared to linear mixed-effects models. As a result, we encountered more frequent convergence problems and singularity warnings when fitting GLMMs for accuracy than when fitting linear mixed-effects models for reaction time, necessitating simpler or alternative model structures. We have added the following sentence to the Methods section (Statistical analysis) on page 17 to clarify this point:

“As generalized linear mixed-effects models for accuracy were more prone to convergence issues and singularity warnings than linear mixed-effects models for reaction time, simpler model structures were chosen.”

Comment 5

To increase readability, I suggest to add brief introduction or definition of the following terms in the Results section when they were first mentioned: a main effect of session, category difficulty, executive abilities, etc.

Reply

We have added a brief description of the respective predictor to the Results section where relevant. For instance, for executive abilities we added the following sentence on page 5:

“Executive abilities were based on an average score of the Trail Making Test (TMT; Reitan, 1958) and the Digit Symbol Substitution Test (DSST; Wechsler, 1944).”

And for the measure of verbal intelligence (Spot-the-Word test) on page 7:

“Results revealed higher accuracy [...] with higher scores in the Spot-the-Word test (STW; Schmidt & Metzler, 1992), which we used as a measure of verbal intelligence ($OR_{STW} = 1.20$, $p = 0.035$, Fig. S3c).

Comment 6

Given TMS's limited spatial focality (as also claimed by the authors), could stimulation targeting the left IFG also affect the adjacent posterior MFG?

Reply

We acknowledge that, due to TMS's limited spatial focality, some induced electric fields may have spread into adjacent regions, including the posterior middle frontal gyrus (MFG). To directly address this concern, we have now included the posterior MFG as an off-target control region in our updated e-field modeling (see Fig. 1c and d). The region of the posterior MFG was defined based on subregions of the Human Brainnetome Atlas (Fan et al., 2016, see Fig. 1b for the ROI). The results show that the induced field strength in the posterior MFG was significantly lower than in our stimulation targets, left IFG and pre-SMA, supporting the spatial specificity of our primary stimulation target. Furthermore, correlation and multiple regression analyses comparing e-fields in the target regions versus the posterior MFG confirmed that behavioral effects were specifically associated with stimulation of the IFG and pre-SMA, not the posterior MFG. These findings support the regional specificity of our stimulation protocol, despite the inherent spatial limitations of TMS.

Figure 1. Task-specific performance disruptions induced by the electric field (e-field) strength. **Panel a** shows the induced e-fields over our stimulation targets in the left IFG and pre-SMA. **Panel b** displays significant correlations of e-fields with behavior after FDR correction. Stronger e-fields in the IFG were associated with slower reactions during semantic fluency, while stronger e-fields in the pre-SMA were linked to poorer performance in figural fluency. **Panel c** shows the e-field strength in subregions of our stimulation targets, anterior (BA45) and posterior (BA44) IFG, and left and right pre-SMA, and in an adjacent off-target region, the left posterior middle frontal gyrus (MFG). Anatomical parcels are based on cytoarchitectonic probabilistic maps (Julich-Brain Atlas, Human Brainnetome Atlas). E-field strength was significantly stronger in our target site in IFG (BA45) and left pre-SMA compared with neighboring regions. **Panel d** shows correlations between e-fields and behavioral measures. For reaction times (RT)

in semantic fluency, only the e-field in the IFG shows an effect. For accuracy in figural fluency during dual-site stimulation, multiple regression including both the IFG and pre-SMA e-fields reveals a significant effect for the pre-SMA, whereas regression with the MFG and pre-SMA e-fields does not.

We have added the following information to the Methods section (page 16):

“Furthermore, to evaluate off-target stimulation effects, we also included the posterior middle frontal gyrus (MFG) as a comparison region, using a probabilistic parcel from the Human Brainnetome Atlas (Fan et al., 2016). We compared the relationship between the e-field strength in our IFG target and the MFG off-target to explore the spatial specificity of TMS.”

And to the Results section (page 9), and have updated Figure 4 in the manuscript (page 8):

“Finally, we defined an off-target region in the posterior middle frontal gyrus (MFG) to further assess the spatial specificity of the TMS effects targeting the IFG. E-field modeling confirmed that the induced stimulation strength in the posterior MFG was significantly lower than in the targeted IFG ($\beta_{BA45 - MFG} = 14.88, p < 0.001$) and pre-SMA regions ($\beta_{\text{left pre-SMA} - MFG} = 20.59, p < 0.001$, Fig. 4c) during the respective session, supporting the focality of our stimulation protocol. Moreover, unlike the IFG and pre-SMA, the e-field in the posterior MFG showed no relationship with behavioral performance, reinforcing the conclusion that observed behavioral effects are specific to stimulation of the intended target areas (Fig. 4d).”

Comment 7

The left IFG in this study does not contain its orbital part, and the manuscript have not mentioned this point.

Reply

Thank you for this observation. We confirm that the left IFG region targeted in our study did not include the orbital part (pars orbitalis), as our stimulation site was located at the border of Brodmann areas 44 and 45. Consequently, the orbital part was not considered further in our analyses. Moreover, our e-field modeling indicates that TMS-induced effects in the pars orbitalis were minimal. We have added the following statement to the Discussion section (page 13) to clarify this point:

“It should be noted that the anatomical definition of the left IFG in our study specifically excluded the pars orbitalis, as our stimulation site was located at the border of Brodmann areas 44 and

45. This limitation should be considered when interpreting the generalizability of our findings to the broader IFG.”

Reviewers #2 and #3

The paper is well-written and makes an important contribution to the literature on brain regions that support semantic fluency as well as the domain-specificity versus -generality of these processes. Our major concerns relate to better delineating how the tasks measure executive control, given the absence of conditions that manipulate executive control demands, as well as better defining clustering and switching responses that are used to measure executive control processes. Describing these elements will be essential for conveying the theoretical impact of the work. We also have some clarification questions regarding the methods and analyses.

Reply

Thank you for the positive evaluation of our work and your helpful comments.

Major Comments

Comment 1

While semantic fluency may broadly involve executive control, there are also specific conditions that place greater demands on executive control than others. For example, Novick et al. (2009) found that a patient with focal damage to the left IFG was selectively impaired in fluency for larger semantic categories (compared to smaller semantic categories), as the larger categories have greater selection demands due to there being more available correct responses. Similarly, picture naming engages left IFG to a greater extent when a picture is associated with more than one name (Kan & Thompson-Schill, 2004), suggesting greater executive control demands for pictures with low name agreement. While the authors employed tasks thought to involve executive control, they did not explicitly manipulate executive control demands in these tasks, limiting their ability to determine if TMS selectively disrupted conditions hypothesized to rely on executive control. We would like to see more discussion of the executive control demands of each task they used. In the absence of conditions manipulating executive control, how do we know that the TMS-induced disruptions are due to executive/semantic control per se as opposed to a different ability like processing speed that is also shared across semantic and figural fluency tasks?

Reply

Thank you for this thoughtful comment. We agree that executive control demands can vary across and within fluency tasks. In our study, we explicitly modeled these demands: for semantic fluency, we included both “easy” and “difficult” categories, with difficulty empirically determined via pilot data reflecting the number of items generated per category (Martin et al., 2022). This approach is based on two premises: (i) Categories with more exemplars require greater selection and inhibition, thus increasing executive control demands (Novick et al., 2009) and (ii) categories with fewer exemplars can make them more demanding over time as retrieval rates decline (e.g., Troyer et al., 1997). For figural fluency, we piloted and matched dot configurations to control for item difficulty and potential executive load.

We accounted for these item-related differences by including category difficulty and stimulus list as predictors in our mixed-effects models. Our results showed that category difficulty robustly predicted performance in semantic fluency, indicating that our manipulation successfully captured variation in executive control demand. However, we did not observe a significant interaction between executive control demand and stimulation condition, suggesting that TMS effects were not limited to the most demanding items. Instead, TMS-induced disruptions appeared to affect overall task performance, aligning with our hypotheses and previous findings regarding the role of the IFG and pre-SMA in semantic and figural fluency, respectively.

While our tasks did not include a separate, orthogonal manipulation of executive control (e.g., through explicit selection or inhibition tasks), our approach allowed us to model and test the impact of varying executive demands within each fluency task. Thus, we interpret the TMS effects as reflecting disruption of executive/semantic control processes, rather than general processing speed, though we acknowledge this as a limitation and an area for future research. We have added the following paragraph to the Discussion section (pages 12-13) to acknowledge this important point:

“In this study, we accounted for variation in executive control demands by including empirically defined “easy” and “difficult” categories in the semantic fluency task and by piloting and matching dot configurations for figural fluency. Although category difficulty robustly predicted performance, we did not observe a significant interaction between executive control demand and stimulation condition, indicating that TMS effects were not limited to the most demanding items. These findings suggest that TMS-induced disruptions affected overall task performance, supporting the involvement of the IFG and pre-SMA in domain-general executive processes.”

Comment 2

The authors examined clustering and switching of responses during verbal fluency to understand executive control effects. Given that executive control demands were not manipulated through task conditions, this analysis appears to be central to the research question of how TMS to left IFG/pre-SMA affects executive control during fluency. To this end, we would like to see more detail about what constituted clustering versus switching. How are these defined, both conceptually and operationally? How does clustering versus switching index executive control processes?

Reply

Clustering refers to the production of words within a semantic or phonemic subcategory, such as listing several farm animals in a row when asked to name animals. **Switching** is defined as the ability to shift from one subcategory (cluster) to another, for example, moving from farm animals to zoo animals during the task. Operationally, we employed FastText (Bojanowski et al., 2017) for fully automated switch detection. This method estimates the number of switches among clusters by calculating the cosine similarity between the vector of any uttered word and the subsequent word vector. If the cosine similarity falls below a predefined switch threshold, a new cluster is identified. We defined category-specific thresholds as the median similarity across all word pairs within a given semantic category as suggested by Alacam et al. (2022). We thus counted a cluster as two or more consecutive words above the predefined switch threshold and a switch each time a participant transitioned from one cluster to a different subcategory or produced a word that did not belong to the previous cluster. While clustering is thought to reflect the integrity of semantic memory and temporal lobe processes (e.g., Pagliarin et al., 2021; Troyer et al., 1997), switching indexes executive control abilities, specifically, cognitive flexibility and strategic search, which are supported by frontal lobe function (e.g., Bose et al., 2016; Troyer et al., 1997) and were thus hypothesized to be affected by TMS.

We added the following paragraph to the Methods section (*Exploratory quantitative analysis for semantic fluency*) on page 18 to better clarify this analysis:

[...] We thus counted a cluster as two or more consecutive words above the predefined switch threshold and a switch each time a participant transitioned from one cluster to a different subcategory or produced a word that did not belong to the previous cluster. While clustering is thought to reflect the integrity of semantic memory and temporal lobe processes (Troyer et al., 1997), switching indexes executive control abilities, specifically, cognitive flexibility and strategic search, which are supported by frontal lobe function (Troyer et al., 1997) and were thus

hypothesized to be affected by TMS (see Supplementary Materials for a more detailed description regarding the definition of clustering and switching).

Additionally, we have added a more detailed description of the definition of clusters and switches as described above to the Supplementary Materials.

Minor Comments

Comment 3

In 64-67 - While the paper notes that participants completed four sessions with different stimulation conditions (IFG, pre-SMA, dual-site, sham), it is not immediately clear that each stimulation condition occurred in a separate session and that the dual-site stimulation (IFG followed by pre-SMA) was administered within a single session. Making this distinction more explicit here or in the figures would improve clarity.

Reply

We have rewritten these lines as follows to clarify the experimental design (page 3):

“In this study, we used transcranial magnetic stimulation (TMS) to investigate the causal roles of the pre-SMA and left anterior IFG in semantic-specific and domain-general executive control. Across four sessions, we applied offline repetitive TMS (rTMS) either to the left anterior IFG alone, to the pre-SMA alone, or subsequently to both regions using a dual-site stimulation approach; a fourth session with sham stimulation served as a control. After each stimulation session, participants completed semantic and figural fluency tasks (see Fig. 1).”

Comment 4

In 99 – “TMS over the left IFG and Pre-SMA both delay Semantic and Figure Fluency” -> It seems like “both” is in the wrong place. Consider either “TMS over both the left IFG and Pre-SMA” or “delay both Semantic and Figure Fluency.”

Reply

Thank you for this careful observation. We have rewritten the sub-heading on page 4 to:

“TMS over the left IFG and Pre-SMA delays both Semantic and Figural Fluency”

Comment 5

In 136-137 & Figure 2C – Why is the TMS-related delay being compared to dual-site rather than sham stimulation for picture naming?

Reply

Thank you for your question. In our analyses, sham stimulation was set as the reference level for the stimulation condition factor, so all primary model comparisons were made against sham by default. However, to fully explore differences between all stimulation conditions, we conducted post-hoc pairwise comparisons using the emmeans package. For picture naming, although no significant differences were found between effective stimulation and sham, we observed a significant delay in responses after single-site IFG stimulation compared to dual-site stimulation (after multiple comparison correction), which we reported in the results (see lines 134–138 and Figure 2C):

“Analyzing RTs in **picture naming** showed **no significant effect of effective relative to sham rTMS** ($\beta_{\text{IFG}} = 0.01$, $p = 0.381$, $\beta_{\text{Pre-SMA}} = -0.01$, $p = 0.203$, $\beta_{\text{Dual site}} = -0.01$, $p = 0.185$, Fig. 2c, Table S1). However, **post-hoc tests** revealed significantly slower reactions after single-site stimulation to left IFG compared to dual-site stimulation ($\beta_{\text{IFG - Dual-site rTMS}} = 0.02$, $p = 0.046$), with an average latency of 21 ms.”

Comment 6

In 138-140 – “...there was a main effect for trial index with slower reactions with progressing time... and for executive abilities..., which reduced reaction times” -> I initially misinterpreted this sentence to mean that slower reactions occurred for executive abilities (in addition to progressing time), until I got to the clause “which reduced reaction times,” indicating that executive abilities were associated with faster reactions. Consider rephrasing.

Reply

We have rephrased the sentence on page 5 as follows:

“Additionally, we observed a main effect of trial index, with reaction times becoming slower as the task progressed (trial index = 0.0003, $p = 0.003$; Fig. S2). There was also a main effect of executive abilities, such that higher executive abilities were associated with faster reaction times (executive abilities = -0.06, $p = 0.032$; Fig. S2).”

Comment 7

In 172-174 – “poorer performance in figural fluency with increasing intensity of the e-field in pre-SMA ($r = -0.55$, $p = 0.043$, Fig. 4b).” -> Since the preceding sentence was talking about RTs in semantic fluency, it was not immediately clear that “poorer performance” here is referring to accuracy (i.e., slower response times might also be interpreted as poorer performance). Also, the r -value printed on Fig. 4b is -0.56 as opposed to -0.55 stated in the text.

Reply

Thank you for this careful observation. We have adapted the sentence on page 8 as follows:

“Moreover, linear regression combining both e-fields for sessions with dual-site stimulation showed poorer accuracy in figural fluency with increasing intensity of the e-field in pre-SMA (std. beta = -0.56 , $p = 0.043$, Fig. 4b).”

Comment 8

In 300-308 – Was a power analysis conducted to determine the sample size? The authors state that this is a “relatively large sample” given “recent methodological considerations for TMS studies in cognition,” but it is unclear what these methodological considerations are or what their statistical power for effects of interest is.

Reply

We did not conduct an a priori power analysis due to limited prior data for effect size estimation. However, our sample size aligns with recent recommendations in meta-analysis on TMS in the modulation of semantic control from Ambrosini et al. (2024), who suggest samples of ≥ 24 participants and the use of mixed-effects models to address inter-individual variability in TMS studies of cognition, supporting more robust statistical analyses.

Comment 9

In 342 – Picture naming lists were balanced for several factors. As items with low name agreement and/or low frequency may increase executive control demands on this “low-level control task,” it would be useful to report some descriptive stats (M , SD or range) about these factors to demonstrate that picture naming was indeed a “low-level control task.”

Reply

Thank you for this helpful suggestion. We have now included descriptive statistics (mean, standard deviation, and range) for word frequency, word length, visual complexity, and name agreement for each of the four picture naming lists to the Supplementary Material, as shown in Table 1 and Figure 2 below. Each list consisted of 99 pictures and was carefully balanced across these variables. Additionally, all images from the MultiPic database are standardized for high concreteness, high name agreement, and high visual clarity, as described by the original authors (Dunabeitia et al., 2018). Based on these characteristics, we are confident that the picture naming task imposed minimal executive control demands for our healthy young participants, especially in comparison to the fluency tasks.

Table 1. Descriptive statistics of stimulus lists for picture naming

Stimulus list	Frequency	Word length in syllables	Name agreement	Visual complexity
1	13.53 (2.14)	2.09 (0.74)	0.64 (0.58)	2.75 (0.55)
2	13.39 (2.18)	2.09 (0.74)	0.61 (0.56)	2.75 (0.53)
3	13.51 (2.31)	2.09 (0.74)	0.63 (0.51)	2.78 (0.54)
4	13.43 (2.30)	2.09 (0.74)	0.61 (0.57)	2.73 (0.52)

Note. For each parameter, mean and standard deviation are given.

Figure 2. Descriptive statistics of stimulus lists for picture naming

Comment 10

In 364-365 – What software/tools were used to convert MNI coordinates into native space?

Reply

We used SPM12 and in-house Matlab scripts to convert coordinates from MNI into native space. The scripts are also available in our publicly accessible repository on OSF: <https://osf.io/q5kam/files/osfstorage>

Comment 11

In 385-386 – What was the inter-rater reliability for RTs that were identified using Praat?

Reply

Thank you for your question. Given the large number of semantic fluency recordings (24 participants × 4 sessions × 6 categories, each 90 seconds), we did not perform formal inter-rater

reliability assessments for reaction times. However, all raters underwent comprehensive training on a subset of recordings, and their judgments regarding word onset and offset were reviewed by multiple researchers to ensure consistency in RT identification.

Comment 12

In 400-401 – “Linear regression models were applied to log-transformed RTs” -> Why transform RTs to meet assumptions of linear regression rather than using GLMMs on the raw RT data (as recommended by Lo & Andrews, 2015)?

Lo S and Andrews S (2015) To transform or not to transform: using generalized linear mixed models to analyse reaction time data. Front. Psychol. 6:1171. doi: 10.3389/fpsyg.2015.01171

Reply

Thank you for raising this point. While GLMMs have been recommended for analyzing raw, skewed RT data, in our dataset, log-transforming RTs and analyzing them with LMMs provided well-behaved, normally distributed residuals and allowed us to flexibly specify complex random effects structures. In contrast, we found that GLMMs (e.g., using a gamma distribution) were less flexible regarding random effects and sometimes encountered convergence issues with our data. Thus, we opted for the log-transformation approach, which is widely used and appropriate when it achieves model assumptions and interpretability for the research question (Baayen & Milin, 2010).

Comment 13

In 401-403 – “Accuracy data were analyzed using logistic regression for the binomially distributed data of figural fluency and picture naming, and negative binomial regression for the count data of semantic fluency.” -> Why isn't figural fluency measured as count data as well? As a fluency task, it seems like the relevant metric is the number of unique figures produced in the allowed time.

Reply

Thank you for your question. We analyzed figural fluency using binomial regression rather than a count model because the number of possible unique figures was capped by the task design,

creating an upper limit on the number of correct responses. Count models like Poisson or negative binomial regression assume no upper bound for the outcome variable, which was not the case here. Therefore, binomial regression was more appropriate for accurately modeling the bounded nature of the figural fluency data.

Comment 14

In 419-420 – “Where possible RT models also included task-specific by-participant random slopes.” -> What does ‘where possible’ mean? Is this determined by the design, by model convergence, or some other factor? Also, why were random slopes not considered for accuracy data (Table 1) or ROI e-field values (In 381-382). Not including random slopes can bias estimates (Barr et al., 2013).

Barr, D. J., Levy, R., Scheepers, C., & Tily, H. J. (2013). Random effects structure for confirmatory hypothesis testing: Keep it maximal. Journal of Memory and Language, 68(3), 255–278. <https://doi.org/10.1016/j.jml.2012.11.001>

Reply

Thank you for raising this important point. We agree that, in principle, maximal random effects structures, including random slopes, are recommended to avoid bias in mixed-effects modeling (Barr et al., 2013). In practice, however, the inclusion of random slopes is often constrained by the variability present in the data and by model convergence issues. For example, in tasks like picture naming, where performance was near ceiling for most participants, there was insufficient variability to support reliable estimation of random slopes. Additionally, GLMMs for accuracy data often proved less flexible and more prone to convergence or singularity problems when random slopes were included. Therefore, the phrase “where possible” refers to instances where the data supported the inclusion of random slopes without causing convergence or singularity warnings; in other cases, simpler random structures were necessary to achieve reliable model estimates.

However, we have now updated the LMM for the ROI e-field values to include by-participant random slopes for session. This adjustment did not change the results, which are presented in the updated Table S4 in the Supplementary Material. We have also revised the relevant statement in the Methods section (Transcranial Magnetic Stimulation) on page 16 to accurately reflect this change:

“A linear mixed-effects model with ROI and session (stimulation target IFG or pre-SMA) as fixed effects and random intercepts for participants as well as by-participant random slopes for session was used to compare e-field values in the ROIs.”

Moreover, reviewer #1 raised a similar question in comment 4. To clarify the decision of our model set up, we have added the following sentence to the Methods section (Statistical analysis) on page 17 to clarify this point:

“As generalized linear mixed-effects models for accuracy were more prone to convergence issues and singularity warnings than linear mixed-effects models for reaction time, simpler model structures were chosen.”

Comment 15

Figure S2a – Why do the condition means for RT by each trial type appear to be less than or equal to the minimum participant RTs?

Reply

Thank you for this insightful question. The condition means for reaction times shown in Figure S2a are estimated marginal means derived from our statistical model using the emmeans package (Lenth, 2024). These values represent model-based predictions that adjust for other variables and covariates in the model, rather than simple raw averages. Because the model accounts for factors such as participant variability and additional predictors, the estimated means can sometimes be lower than the observed raw data means. This is expected, as the estimated marginal means reflect the predicted response for each category type while controlling for other influences, providing a more accurate representation of the isolated effect of category type. In contrast, the individual data points in the background represent raw data averaged by participant, which do not account for these model adjustments and therefore may not align directly with the model-based predictions.

Response Letter

We thank all reviewers for their positive feedback and constructive comments. We have addressed each point in detail below and believe that these revisions have substantially improved the quality and clarity of the manuscript, making it well-suited for publication.

Reviewer #1

Comment 1

The newly added analyses of E-Field strength in left posterior MFG reported in Figure 4 are interesting. However, the effect size of relationship of E-Field strength and behavioral performance was comparable between the left posterior MFG and the left IFG ($r = 0.38$ vs. 0.42 ; and the positive correlation was marginally significant for the left posterior MFG). I suggest that the authors provide additional evidence demonstrating that TMS selectively disrupts left IFG function while leave MFG function intact.

Reply

Thank you for your thoughtful comments regarding the e-field analysis. As you noted, the e-field strength in the posterior MFG is similar to that in the IFG, both in magnitude and direction. This is an expected outcome given the spatial spread of TMS-induced fields in the cortex (Numssen et al., 2023). Our simulations show that the strongest e-field is induced in our target region, the IFG (BA45), with statistically significant differences after multiple comparison correction compared to adjacent regions (BA44 and posterior MFG, see Fig. 4c). While the correlation between e-field strength and behavioral performance in the MFG is somewhat similar to that in the IFG, only the IFG effect reaches statistical significance after correction for multiple comparisons. Consistent with best practices (Olsson-Collentine et al., 2019; Pritschet et al., 2016), we avoid interpreting marginal p-values that do not meet our pre-specified threshold for significance ($p = 0.05$). Importantly, the primary aim of our study was not to directly contrast TMS effects in IFG and MFG, but rather to compare IFG stimulation to pre-SMA, which our findings address. Nevertheless, we agree this is a promising direction for future research, which may further clarify the distinct contributions of middle and inferior frontal gyrus regions to cognitive control.

To acknowledge the observed effect in MFG, we have included the following statement in the Discussion (p. 13):

“Moreover, the post-hoc e-field simulation in the posterior MFG showed similar strength and direction to the e-field in our target region IFG, though statistically with reduced strength and not the same relationship with behavioral performance. Future research is needed to further disentangle the contribution of IFG and posterior MFG to different domains of cognitive control.”

Comment 2

Table S2. It is counterintuitive that TMS over either IFG or PreSMA individually affected reaction times in the figural fluency task, whereas concurrent TMS over both regions did not. I suggest adding possible explanations for this result.

Reply

Thank you for highlighting this result. In our study, dual-site TMS was administered sequentially within the same session rather than simultaneously. The slightly reduced effect size following subsequent dual-site stimulation possibly reflects short-term neural adaptation. In this context, network state changes induced by TMS over the IFG may have diminished the response to subsequent TMS over the pre-SMA, leading to a non-additive behavioral outcome. This underscores the importance of considering order and temporal effects in dual-site TMS designs.

Reviewer #2

Minor Comments

Comment 1

p. 5 – The authors now state that executive ability was measured as the average of scores on TMT and DSST. On p. 17, the authors clarify that both metrics were converted to z-scores prior to averaging. This should be specified here as well to avoid confusion.

Reply

Thank you for this constructive comment. We have now added this information to the sentence on p. 6, which now reads as follows:

“Executive abilities were based on an average score of z-standardized scores of the Trail Making Test (TMT; Reitan, 1958) and the Digit Symbol Substitution Test (DSST; Wechsler, 1944).”

Comment 2

p. 9 – There is still a mismatch between the regression coefficient between e-field strength of the pre-SMA and accuracy in figural fluency reported in the text ($r = -0.55$) and Figure 4b ($r = -0.56$). The number was updated in the rebuttal letter but not in the main text.

Reply

Thank you for your careful review. We have now corrected the number in the manuscript on p. 8 to read $(r = -0.56)$.

Comment 3

p. 13 – “Although category difficulty robustly predicted performance, we did not observe a significant interaction between executive control demand and stimulation condition, indicating that TMS effects were not limited to the most demanding items.” -> As far as we can tell, the interaction between category difficulty and stimulation condition wasn’t tested: the models for semantic fluency in Table 1 indicate fixed effects of stimulation condition and category difficulty but not their interaction. The description of statistical analyses on p. 17-18 mentions that stepwise model selection was performed, but never mentions interactions being included as part of the model testing. If interactions were tested, this needs to be stated. Otherwise, it is not appropriate to conclude that TMS effects applied to all semantic fluency items, regardless of whether they were “easy” or “hard.”

Reply

We apologize for this oversight. We tested for an interaction of stimulation condition and category type based on your comments in the previous review round but did not include the results in the manuscript. We have now added the following statement to the Results section (p. 6):

“We also tested for a potential interaction of stimulation condition and category type, hypothesizing that TMS may selectively show stronger disruptions for more demanding semantic categories. However, the interaction was not significant ($X_{\text{Stimulation condition:Category type}} = 0.70$, $p = 0.873$) and reduced overall model performance compared to the original LMM without the interaction term present ($AIC_{\text{SemFluency RT Orig}} = 37821$, $AIC_{\text{SemFluency RT Int}} = 37837$).”

Comment 4

p. 16 – The authors’ response clarifies that SPM12 was used to convert MNI coordinates into native space, but SPM12 is never cited in the text or reference list.

Reply

We apologize for this oversight. We have added the following sentence to the Methods section (p. 17):

“Anatomical data were preprocessed using SPM12 (Wellcome Trust Centre for Neuroimaging) in Matlab (version R2022b).”

Comment 5

Throughout – We appreciate the clarification regarding clustering and switching during verbal fluency in the revised manuscript. Nevertheless, a clear definition of clustering and switching processes is not provided until the Methods on p. 18, well after the results of this analysis are reported (p. 9-10). I recommend defining these terms when they are first introduced on p. 3 of the Introduction. The authors could consider moving the clear descriptions that they provide in the Supplementary Materials (e.g., “Clustering refers to the production of words within a semantic or phonemic subcategory, such as listing several farm animals in a row when asked to name animals. Switching is defined as the ability to shift from one subcategory (cluster) to another, for example, moving from farm animals to zoo animals during the task”) to just after the sentence “Additionally, we employed a novel machine learning-based clustering and switching analysis to examine TMS effects on category switching during verbal fluency, a process highly sensitive to the detection of neurodegenerative diseases (Troyer et al., 1998)” on p. 3. Moreover, this would allow the removal of this section from the Supplementary Materials, as the rest of the information in that section is provided elsewhere in the text.

Reply

Thank you for this comment. Following your suggestion, we have added the following sentence to the Introduction (p. 3) and removed the supplementary information on this analysis:

“Clustering refers to the production of words within a semantic or phonemic subcategory, such as listing several farm animals in a row when asked to name animals. Switching is defined as the ability to shift from one subcategory (cluster) to another, for example, moving from farm animals to zoo animals during the task.”

Comment 6

Table S1 & Figure S2 – The scales of measurement for word frequency and visual complexity need to be clarified.

Reply

Thank you for this careful observation. We have added the information to the Supplementary Materials (Table S1 and Figure S2) on page 31 and additionally added the source of word frequencies to the note of Table S1:

“Word frequencies in the form of frequency class were derived from the database Wortschatz Leipzig (https://corpora.uni-leipzig.de/de?corpusId=deu_news_2022).”